# Towards Understanding Domain Adapted Sentence Embeddings for Document Retrieval

## Abstract

A plethora of sentence embedding models makes it challenging to choose one, especially for technical domains rich with specialized vocabulary. In this work, we domain adapt embeddings using telecom, health and science datasets for question answering. We evaluate embeddings obtained from publicly available models and their domain-adapted variants, on both point retrieval accuracies, as well as their (95%) confidence intervals. We establish a systematic method to obtain thresholds for similarity scores for different embeddings. As expected, we observe that fine-tuning improves mean bootstrapped accuracies. We also observe that it results in tighter confidence intervals, which further improve when pre-training is preceded by fine-tuning. We introduce metrics which measure the distributional overlaps of top-$K$, correct and random document similarities with the question. Further, we show that these metrics are correlated with retrieval accuracy and similarity thresholds. Recent literature shows conflicting effects of isotropy on retrieval accuracies. Our experiments establish that the isotropy of embeddings (as measured by two independent state-of-the-art isotropy metric definitions) is poorly correlated with retrieval performance. We show that embeddings for domain-specific sentences have little overlap with those for domain-agnostic ones, and fine-tuning moves them further apart. Based on our results, we provide recommendations for use of our methodology and metrics by researchers and practitioners.

## 1 Introduction

Document Question Answering (QA) methods such as Retrieval Augmented Generation (RAG) typically involve retrieval of sections, paragraphs or sentences from a document corpus to accurately answer user queries. Embedding models are used to map the questions or documents to a semantic space. Retrieval is typically achieved by computing similarity between embeddings of questions and those of documents. The most similar top-$K$ documents are considered to be the relevant answers.

While many state-of-the-art (SOTA) models trained on publicly available datasets are easily accessible (Reimers & Gurevych, 2019; Chen et al., 2024; Zhang et al., 2023; Xiao et al., 2023), obtaining good retrieval accuracies for domain specific tasks is challenging (Roychowdhury et al., 2024). It is well acknowledged in literature that domain adaptation and fine-tuning can improve retrieval performance (Li et al., 2020), but making an informed choice among several available models involves extensive evaluation over parameters such as the number of relevant documents retrieved for a test set.

Some studies (Zhou et al., 2022) have identified limitations of cosine similarities to retrieve embeddings - a sample limitation is underestimation of similarity of frequent words with their homonyms. It has been shown that cosine similarities can be arbitrary or dependent on regularization, making them unreliable for retrieval tasks (Steck et al., 2024) - although this study was limited to linear models the authors have conjectured that the same may be true for non-linear models. In fact, variations in embedding space representations obtained from different architectures have been widely studied (Mistry & Minai, 2023; Biś et al., 2021; Timkey & Van Schijndel, 2021). Another limitation observed is reporting of point accuracies, without any error bars, for retrieval tasks. This limits estimation of performance on new questions, especially when evaluated with relatively small datasets.

Recent work has explored isotropy as a measure for quantifying robust embedding space representations (Jung et al., 2023; Rudman & Eickhoff, 2023; Rudman et al., 2021), though it has also been argued otherwise (Hou et al., 2024; Ait-Saada & Nadif, 2023; Godey et al., 2023; Razzhigaev et al.,

2023). In particular, Jung et al. (2023) suggests that isotropic embeddings improve retrieval whereas Rudman & Eickhoff (2023) propose that reduced isotropy or anisotropy helps retrieval. Rajaee & Pilehvar (2021) looks at isotropy of embeddings and show that increasing the isotropy of fine-tuned models leads to poorer performance.

We observe a few limitations with the current practice of measuring retrieval performance in both research and practice. First, reporting point accuracies do not provide insight into error bars (confidence intervals). This is especially important for relatively smaller datasets. Second, the lack of confidence intervals does not allow for tests of statistical significance when comparing different embedding models or domain adaptation strategies. Third, to the best of our knowledge, we have not found prior work which has provided a systematic approach to choose the best threshold. In practice, such thresholds are often chosen by inspection of similarity scores. Our approach of bootstrapping provides the ability to perform tests for statistical significance on the results, and we choose the maximum threshold such that our results are not statistically worse off. Finally, although prior work Gao et al. (2021); Ethayarajh (2019) have looked at the effect of domain adaptation on embeddings, the separation of domain-specific embeddings from general purpose embeddings under domain adaptation has not been studied. This does not allow a clear understanding of why performance changes on general purpose retrieval post domain adaptation.

## 1.1 RESEARCH QUESTIONS AND CONTRIBUTIONS

The primary research questions in this work are:

- **RQ1**: What are the confidence intervals (CI) of accuracies of SOTA retrieval models and their fine-tuned versions when considering domain specific tasks?

- **RQ2**: What facets apart from retrieval accuracies can characterise an embedding model? How does the distribution of cosine similarities vary across embeddings?

- **RQ3**: Can the variation of retrieval accuracies be attributed to only the isotropy of the embeddings?

Our **primary contributions** are:

- Demonstrate that fine-tuning improves mean accuracies as well as CI. Pre-training followed by fine-tuning improves CI further.

- Propose a systematic method to introduce thresholds with minimal effect on retrieval accuracies.

- Show that although domain adaptation via fine tuning leads to higher isotropy scores, retrieval performance across models is poorly correlated with the isotropy scores of the models; improving isotropy scores via transformations does not improve accuracies.

- We introduce metrics which measure the distributional overlaps of top-$K$, correct and random document similarities with the question.

- Show empirically that these metrics are correlated with accuracies and similarity thresholds.

- Demonstrate that domain adaptation shifts the embeddings of the target domain farther away from embeddings of sentences from domain agnostic datasets.

The rest of the paper is structured as follows: the methodology is detailed in Section 2. We detail the domains considered along with the corresponding datasets and models in Section 3.1 and Section 3.2 respectively. We report experimental results of multiple embeddings (with and without domain adaptation) in Section 4. We summarize our findings and discuss limitations and scope for future work in Section 5.

## 2 METHODOLOGY

We consider the following in this study *viz.* computing bootstrapped accuracies, estimating probabilities of overlap between different distributions, analysis of minimum thresholds for similarities and study the effects of isotropy scores. We describe each of these formally in this section. For most

of our experiments, we choose a bootstrapped approach to get both point estimates and CI for our estimates.

Consider a dataset $\mathcal{D} = [s_1, s_2, \ldots, s_N]$, where $s_i$ is the $i^{th}$ sentence and $i \in [1, N]$. Let $\mathcal{D}$ be associated with a question set $\mathcal{Q}$, containing $Q$ questions. Each question $q \in \mathcal{Q}$ can be uniquely answerable by one sentence $s_q \in \mathcal{D}$, which we consider as the correct answer for the question $q$. Let the embedding representation of $s_i$ using a sentence embedding model $\mathcal{M}$ be represented by $E_{\mathcal{M}}(s_i)$, and correspond to dimension $\mathcal{M}_p$. Similarly, let $E_{\mathcal{M}}(q)$ represent the embedding (using sentence embedding model $\mathcal{M}$) for a question, $q \in \mathcal{Q}$. Henceforth, in this work, all sentence embeddings will be referred to as embeddings.

Like in any typical QA retrieval methodology, $\mathcal{D}$ and $\mathcal{Q}$ result in embedding matrices of sizes $N \times \mathcal{M}_p$ and $Q \times \mathcal{M}_p$ respectively. All embeddings are normalized to have unit $L_2$ norm. We draw $m$ bootstrap samples from $\mathcal{Q}$, each containing $l$ questions i.e., $|\mathcal{Q}_j| = l$ with $|\cdot|$ indicative of the cardinality of the corresponding set and $j \in [1, m]$. We use these bootstrapped samples in our experiments.

## 2.1 BOOTSTRAPPED ACCURACIES

Consider any $j^{th}$ bootstrap sample $\mathcal{Q}_j \in \mathcal{Q}$. For each question $q \in \mathcal{Q}_j$, we find the set $t_q^K$ of the top-$K$ most similar sentences based on highest cosine similarity and check if $s_q$ is included in this set. The top-$K$ accuracy, $a_j$, is the proportion of questions in this bootstrap sample for which $s_q \in t_q^K$. The mean bootstrapped retrieval accuracy is given by $a = \frac{1}{m} \sum_{j=1}^{m} a_j$.

The 95% confidence interval $(a_{\text{lower}}, a_{\text{upper}})$ is defined by the $2.5^{th}$ and $97.5^{th}$ percentiles of the set of $a_i$ values.

## 2.2 COMPUTATION OF THRESHOLDS

It is often desirable to have thresholds on similarity scores between questions embeddings and retrieved sentence embeddings from the dataset via top-$K$ similarity scores, thus ignoring any sentence with similarity score below this threshold. This reduces retrieval of sentences that may not necessarily answer the question. A low threshold runs the risk of including wrong/irrelevant documents in retrieval results, and a high threshold can reduce the top-$K$ accuracy.

However, there is no reliable way to estimate a threshold, given that the distribution of similarities can be different based on choice of the embedding model. Hence, we follow a bootstrapped analysis.

Consider each of the bootstrap samples, $\mathcal{Q}_j$. We construct a similarity matrix $S_{\mathcal{M}}^j = E_{\mathcal{M}}(\mathcal{Q}_j) \cdot E_{\mathcal{M}}(\mathcal{D})^T$, where $(\cdot)$ denotes the dot product, $()^T$ denotes the matrix transpose and $S_{\mathcal{M}}^j \in \mathbb{R}^{(l \times N)}$. Let $T_{\mathcal{M}}^j$ be constructed such that, each row of $T_{\mathcal{M}}^j$ has the top-$K$ similarity scores from $S_{\mathcal{M}}^j$. We define $\gamma^j = min(T_{\mathcal{M}}^j)$ and $\Gamma \triangleq \{\gamma^j : j \in [1, m]\}$.

Let us choose a threshold, using $\psi^{th}$ percentile of $\Gamma$, defined by $\tau(\psi)$ s.t. $P_{\Gamma}(x < \tau(\psi)) = \psi$. We study the effect of $\tau(\psi)$ on bootstrapped retrieval accuracies. We substitute all similarities of $S_{\mathcal{M}}^j < \tau(\psi)$ to be zero. We consider the threshold as the highest $\tau(\psi)$ such that the accuracy from this substitution is not statistically different from the mean bootstrap accuracy, $a$ (refer Section 2.1). We would like to clarify that $\gamma^j$ is the set of minimum similarities in the bootstrapped samples, thus $\psi$ can be interpreted as the percentile of irrelevant documents - however there is no direct interpretation with respect to the total number of documents retrieved.

## 2.3 ANALYSIS OF DISTRIBUTION OF VECTOR EMBEDDINGS

To understand the vector embeddings in the semantic space and their effect on the retrieval performance, we study the distributions of cosine similarities of vector embeddings from selected models. As mentioned earlier, all embeddings have unit $L_2$ norm.

We first consider $\mathcal{Q}$ and estimate the following distributions:

- **Distribution of correct similarity scores** - Let $sim_q^{corr}$ represent the cosine similarity between $E_{\mathcal{M}}(q)$ and $E_{\mathcal{M}}(s_q)$, $\forall q \in \mathcal{Q}$. Let $S_{corr} = \{sim_q^{corr} : q \in \mathcal{Q}\}$ represent the set of correct similarity scores.

- **Distribution of top-k similarity scores** - Let $sim_q^{topK}$ represent cosine similarities between any question and the corresponding top-$K$ retrieved sentences. Let this set be represented by $S_{topK} = \{sim_q^{topK} : q \in \mathcal{Q}\}$.

- **Distribution of random similarity scores** - Let $sim_q^{rand}$ represent the cosine similarity between embedding of any question, $E_{\mathcal{M}}(q)$, $\forall q \in \mathcal{Q}$ and that of a randomly chosen statement $E_{\mathcal{M}}(s_r)$, s.t. $s_r \in \mathcal{D}$. Let this set be represented by $S_{rand} = \{sim_q^{rand} : q \in \mathcal{Q}\}$.

Evidently, $|S_{corr}| = Q$, $|S_{topK}| = KQ$ and $|S_{rand}| = Q$.

We estimate the Empirical Cumulative Distribution Function (ECDF) [1] for each of these sets; let these be represented by $C_{corr}$, $C_{topK}$ and $C_{rand}$ for $S_{corr}$, $S_{topK}$ and $S_{rand}$ respectively.

Consider each bootstrapped sample $\mathcal{Q}_j$. Let $\theta_j$ be defined as the similarity score at the $\psi^{th}$ percentile of the set $S_{topK}$ i.e., $P_{S_{topK}}(sim^{topK} \leq \theta_j) = \psi$. Now, we define the following ECDF estimates:

$$C_{corr}(\theta_j) \triangleq P_{S_{corr}}(sim^{corr} > \theta_j) \tag{1}$$

$$C_{rand}(\theta_j) \triangleq P_{S_{rand}}(sim^{rand} > \theta_j) \tag{2}$$

These are a measure of the overlap of cosine similarities between top-$K$ and correct, top-$K$ and random QA sentence pairs. The mean of these across the bootstrapped samples can be calculated as $\bar{C}_{corr}(\theta) = \frac{1}{m}\sum_{j=1}^m C_{corr}(\theta_j)$ and $\bar{C}_{rand}(\theta) = \frac{1}{m}\sum_{j=1}^m C_{rand}(\theta_j)$. We refer to them as correct-overlap-ECDF (COE) and random-overlap-ECDF (ROE) estimates.

We also estimate the 95% CI for both COE and ROE by the using the $2.5^{th}$ and $97.5^{th}$ percentile of $C_{corr}(\theta_j)$ and $C_{rand}(\theta_j)$ as lower and upper bounds respectively.

### 2.4 DOMAIN ADAPTATION

One of the key challenges in leveraging embedding models for technical domains is the lack of domain specific knowledge, since the SOTA (base) models have been trained on publicly available datasets which may be minimally introduced to domain specific terminology. We evaluate various domain adaptation techniques on the base models:

- Pre-training Li et al. (2020): We use Masked Language Modeling (MLM) (Salazar et al., 2019) approach for this. Sentences from the corpus of technical documents (of a domain) are used.

- Fine-tuning (Mosbach et al., 2020): We prepare triplets of the form $< q, p, n >$ where $q$ corresponds to the user query, $p$ represents the correct (positive) answer and $n$ is a list of incorrect (negative) answers. The base model is fine-tuned using these triplets. It may be noted here that the fine-tuning may be performed post pre-training or independently on the base model (without pre-training).

Thus, we evaluate the following variants of embedding models - base model, pre-trained only (PT), fine-tuned only (FT) and pre-training followed by fine-tuning (PT-FT). As recommended[2], post fine-tuning, we merge the base model with the domain adapted model.

### 2.5 ISOTROPY SCORES

Isotropy measures distribution of embeddings on the high-dimensional unit hyper-sphere (since all embeddings have unit-$L_2$ norm). If the embeddings are uniformly distributed over the unit sphere i.e.

---

[1] https://docs.scipy.org/doc/scipy/reference/generated/scipy.stats.ecdf.html

[2] https://github.com/FlagOpen/FlagEmbedding/blob/master/examples/finetune/README.md

there is no preferred direction, then, they are said to be isotropic Arora et al. (2016); Mu et al. (2017). We use two different measures of isotropy to validate our findings (for details refer to Appendix A.3). We represent the isotropic scores as, $I_A$, the second order approximation as defined in Mu et al. (2017) and $I_B$ to be isoscores as per Rudman et al. (2021; 2022). These measure isotropy differently and thus their scores can be quite different. Higher isotropic scores implies embeddings being well distributed in the unit hyper-sphere.

Various transformations have been proposed in literature to improve the isotropy scores. We choose the following to study the effect of isotropy (measured using both $I_A$, $I_B$) on retrieval accuracies.

- Whitened: Whitening of embeddings (Jung et al., 2023)
- PCA: Post-processing embeddings by centering and eliminating the top principal components (Mu et al., 2017)
- Standardized: Mean subtraction and unit standard deviation (Timkey & Van Schijndel, 2021)

## 2.6 COMPARISON OF EMBEDDINGS POST DOMAIN ADAPTATION

We analyze the effect of pre-training and fine-tuning base embedding models with domain-specific data by comparing distribution of the resultant embeddings with that of embeddings from a domain agnostic dataset.

Let $\mathcal{D}$ represent domain-specific data, $\mathcal{D}'$ represent domain-agnostic dataset. Let $\mathcal{M}$ be the base model, $\mathcal{M}'$ be the pre-trained, fine-tuned version of the base model. Let similarity between the datasets be defined $\Delta_{\mathcal{M}}(\mathcal{D}, \mathcal{D}') \triangleq \{min(||E_{\mathcal{M}}(d), E_{\mathcal{M}}(d')||_2) : d \in \mathcal{D}, d' \in \mathcal{D}'\}$, and $|\Delta_{\mathcal{M}}(\mathcal{D}, \mathcal{D}')| = |\mathcal{D}|$.

We compare the distributions of $\Delta_{\mathcal{M}}$ and $\Delta_{\mathcal{M}'}$. Our motivation here is to analyse the separation of the distributions post domain adaptation.

## 3 EXPERIMENTAL SETUP

### 3.1 DATASETS

For our experiments, we consider three domains - telecom, medical and science and use one dataset from each of them. We also consider one domain agnostic dataset. Table 1 has a brief summary of the size of the train and test splits. Note that, we do not do any pre-training/fine-tuning for the domain agnostic dataset hence train dataset size for this is "Not Applicable" (N/A). The TelecomQuad dataset is propiertary and the citation for the same is masked for blind review.

| Domain | Dataset | Reference | Train Dataset Size | Test Dataset Size |
|---|---|---|---|---|
| Telecom | TelecomQuad | Masked | 4186 | 981 |
| Health | PubMedQA | Jin et al. (2019) | 4000 | 500 |
| Science | Sciq | Johannes Welbl (2017) | 10481 | 884 |
| Domain Agnostic | SQuAD | Rajpurkar et al. (2016) | N/A | 1009 |

Table 1: Summary of Datasets used in our experiments

We choose $K = 5$ for reporting top-$K$ accuracies. For bootstrap experiments, we consider $l = 100$, $m = 500$.

### 3.2 EMBEDDING MODELS

We consider the following embedding models for our experiments:

- From the BAAI family of embedding models, we consider *bge-large-en* (Xiao et al., 2023) and *llm-embedder* (Zhang et al., 2023) with $\mathcal{M}_p = 1024, 768$ respectively. We PT, FT, PT-FT these models for further experiments.
- In addition, only for the telecom dataset
    - We evaluate a telecom-domain adapted BERT-based model General-Telecom-Embeddings (GTE) with $\mathcal{M}_p = 768$.

| Embedding Model | Bootstrapped Acc | CI Width | COE | ROE | $(\tau, \psi)$ | Acc @ $\tau$ |
|---|---|---|---|---|---|---|
| Dataset: TelecomQuAD, k=5 | | | | | | |
| bge_large | 66.87 | 17.04 | 87.98 | 4.81 | 0.5 (35) | 67.18 |
| bge_large_pretrained | 62.64 | 17 | 85.94 | 2.18 | 0.58 (25) | 61.36 |
| bge_large_finetuned | 81.61 | 14.04 | 91.98 | 0.22 | 0.43 (25) | 79.46 |
| bge_large_pretrained_finetuned | 81.67 | 13.04 | 91.06 | 0.23 | 0.4 (35) | 77.73 |
| llm_embedder | 70.06 | 14.52 | 87.26 | 5.77 | 0.78 (30) | 69.9 |
| llm_embedder_pretrained | 57.12 | 19.57 | 84.88 | 6.32 | 0.75 (30) | 52.53 |
| llm_embedder_finetuned | 81.58 | 13.52 | 90.73 | 0.10 | 0.56 (40) | 80.69 |
| llm_embedder_pretrained_finetuned | 80.37 | 12.52 | 90.74 | 0.21 | 0.53 (25) | 77.97 |
| Dataset: sciq, k=5 | | | | | | |
| bge_large | 92.7 | 10 | 96.98 | 4.14 | 0.508 (85) | 92.8 |
| bge_large_pretrained | 92.08 | 9 | 95.48 | 1.99 | 0.5 (15) | 92.1 |
| bge_large_finetuned | 94.45 | 8 | 97.79 | 3.72 | 0.883 (85) | 94.8 |
| bge_large_pretrained_finetuned | 95.5 | 7 | 98.15 | 3.25 | 0.771 (85) | 95.4 |
| llm_embedder | 91.69 | 9 | 97.2 | 6.85 | 0.809 (85) | 91.8 |
| llm_embedder_pretrained | 91.03 | 10 | 94.42 | 2.21 | 0.682 (85) | 91.2 |
| llm_embedder_finetuned | 94.37 | 8 | 98.18 | 7.4 | 0.883 (85) | 94.7 |
| llm_embedder_pretrained_finetuned | 94 | 8 | 98.24 | 6.28 | 0.824 (85) | 93.9 |
| Dataset: PubMedQA, k=5 | | | | | | |
| bge_large | 94 | 9 | 96.2 | 3.21 | 0.426 (5) | 93.8 |
| bge_large_pretrained | 84.3 | 12 | 93.8 | 7.52 | 0.492 (5) | 83.4 |
| bge_large_finetuned | 98.51 | 5 | 99.5 | 4.81 | 0.791 (85) | 98.5 |
| bge_large_pretrained_finetuned | 97.83 | 4 | 98.95 | 6.71 | 0.755 (45) | 97.7 |
| llm_embedder | 95.42 | 8 | 97.7 | 4.49 | 0.732 (5) | 95.8 |
| llm_embedder_pretrained | 91.87 | 10 | 96.94 | 4.1 | 0 (100) | 92.0 |
| llm_embedder_finetuned | 97.53 | 6 | 99.28 | 5.55 | 0.797 (30) | 97.5 |
| llm_embedder_pretrained_finetuned | 97.5 | 5 | 99 | 6.15 | 0.783 (30) | 97.2 |

Table 2: Different metrics for various datasets ($\mathcal{D}$) dataset, $K$=5

- From the sentence transformers (Reimers & Gurevych, 2019) library, we consider MPNET (Song et al., 2020) and MiniLM (*all-MiniLM-L6-v2*). Their $\mathcal{M}_p$ are 768 and 384 respectively.
- From OpenAI family[3], we evaluate on *text-embedding-3-small*, *text-embedding-3-large* and *ada_002*, $\mathcal{M}_p$ being 1536, 3072 and 1536 respectively.

All experiments have been conducted using a A100-SXM4-80GB GPU.

## 4 RESULTS

### 4.1 ACCURACIES AND CONFIDENCE INTERVALS

Table 2 reports retrieval accuracy scores along with confidence intervals across 3 datasets. Here we only present the results for models (BAAI family) which have been domain-adapted. Additional results for telecom domain with public models, custom telecom model and OpenAI models are provided in Appendix A.1.

We observe a consistent accuracy improvements across models and domains on FT and PT-FT. However, we observe that fine-tuning a base model and that of a pre-trained model is not much different from the mean accuracies perspective. More importantly, and to the best of our knowledge not reported previously, is the insight that confidence intervals become tighter with FT and further, with PT-FT. Since only PT is trained with a MLM objective, it is not surprising, and previously observed (Li et al., 2020), that there is a reduction in accuracies for PT models.

We report COE (as defined in Section 2.3) for the various models and domain-specific datasets in Table 2. The correlation between COE and accuracy is reported in Table. 3. We see a strong positive correlation between them across domain-specific datasets.

The column $\tau(\psi)$ in Table 2 indicates the thresholds as per the method described in Section 2.2. While the accuracies have slightly reduced with introduction of thresholds (refer Acc @$\tau$ column), this can be interpreted as the accuracy obtained with removal of less relevant documents in retrieved

[3]https://platform.openai.com/docs/guides/embeddings/embedding-models

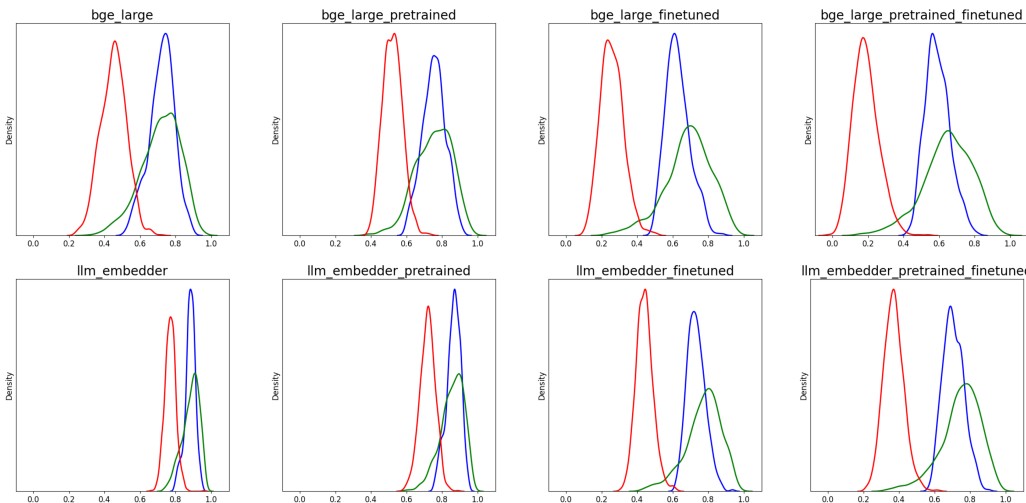

Figure 1: Density plots for telecom dataset. Red, green and blue indicate distribution of $S_{rand}$, $S_{corr}$ and $S_{topK}$ respectively. Refer Sec. 2.3 for definitions

results. Additionally, Acc @$\tau$ **is not statistically different** from the bootstrapped accuracy for the whole dataset (refer column 7 vs column 2). Thus, our choice of threshold **does not lead to degradation of accuracies** in a statistical sense. We repeat here, from Section 2.2, that there is no direct interpretation of $\psi$ with respect to the total number of documents retrieved.

| Corr | Telecom | SciQ | PubMedQA | Average |
|---|---|---|---|---|
| Acc vs. COE | 0.882 | 0.838 | 0.961 | 0.894 |
| Acc vs. ROE | -0.121 | 0.232 | -0.277 | -0.05 |
| Threshold vs ROE | 0.391 | 0.611 | 0.378 | 0.46 |
| Acc vs $I_A$ | 0.014 | -0.082 | -0.255 | -0.108 |
| Acc vs $I_B$ | 0.05 | 0.008 | -0.042 | 0.005 |

Table 3: Correlation values

As expected, the correlation between ROE and accuracy is low (refer Table 3) across domains. We also analyse the correlation between threshold ($\tau(\psi)$) with ROE. This is found to be positively correlated. These correlations are not obvious - this indicates that for a model to perform well, questions must be well interspersed with answers in the embedding space. This is also reflected in the distribution of embeddings as shown in Figure 1. While Figure 1 is for telecom domain, we report the distribution of embeddings for other domains in Appendix A.4.

On further analysing Figure 1, we notice that the *llm_embedder* model has a very peaky distribution of cosine similarities (even for $S_{rand}$). This is indicative of a model with low isotropy. Despite being less isotropic, the retrieval accuracies of the model is similar to the *bge_large* model which is more isotropic. The domain adaptation of *llm_embedder* model creates a wider distribution of the cosine similarities indicating better isotropy. The improvement in isotropy post domain-adaptation has also been reported in Gao et al. (2021).

Retrieval accuracies for SQuAD dataset, $\mathcal{D}'$ are reported in Table 4 using various domain-adapted models. The domain for which the models are trained on is indicated by the merged rows in between. We observe that except *llm_embedder_ptft* on TelecomQuAD, there is a consistent decrease in accuracy of the domain-adapted models. The small improvement in telecomQuAD adapted *llm_embedder_ptft* is perhaps indicative of domain-specific data in the original training of the publicly available models but is hard to quantify. The results in Table 4 demonstrates that domain-adapted models show reduced performance when evaluated on domain-agnostic datasets. NDCG scores are also provided in Appendix A.5

| Embedding Model | Bootstrapped Acc (CI) | $I_A$ | $I_B$, |
|---|---|---|---|
| bge_large | 87.68, (81.0, 93.0) | 13.33 | 45.72 |
| llm_embedder | 86.74, (81.0, 92.0) | 13.63 | 22.88 |
| TelecomQuAD | | | |
| bge_pretrain | 69.31 (60.0, 78.0) | 8,76 | 23.03 |
| bge_ft | 79.21 (69.47, 84.52) | 14.26 | 48.63 |
| bge_pt_ft | 74.23 (66.0, 83.05) | 8.25 | 33.93 |
| llm_embedder_pretrain | 62.93 (54.0, 72.0) | 11.26 | 38.42 |
| llm_embedder_ft | 81.21 (68.47, 85.92) | 14.26 | 48.63 |
| llm_embedder_ptft | 87.25 (80.48, 92.0) | 7.84 | 44.73 |
| sciq | | | |
| bge_pretrain | 76.36 (68.0, 84.0) | 5.46 | 39.54 |
| bge_ft | 84.79 (77.48, 91.0) | 6.69 | 26.84 |
| bge_pt_ft | 83.05 (76.48, 90.0) | 6.21 | 26.12 |
| llm_embedder_pretrain | 77.73 (71.0, 84.0) | 8.53 | 26.07 |
| llm_embedder_ft | 82.23 (75.0, 89.0) | 10.45 | 20.23 |
| llm_embedder_ptft | 85.23 (78.0, 92.0) | 11.91 | 21.82 |
| PubMedQA | | | |
| bge_pretrain | 70.38 (62.0, 79.0) | 6.38 | 35.84 |
| bge_ft | 84.85 (78.0, 91.0) | 8.32 | 29.08 |
| bge_pt_ft | 83.28 (76.0, 89.52) | 7.22 | 30.42 |
| llm_embedder_pretrain | 80.84 (74.0, 87.52) | 9.20 | 25.42 |
| llm_embedder_ft | 83.47 (77.0, 90.0) | 11.38 | 21.95 |
| llm_embedder_ptft | 81.18 (74.0, 88.0) | 9.04 | 22.60 |

Table 4: Accuracies (k=5) and isotropy scores for SQuAD dataset ($\mathcal{D}'$) Intermediate rows indicate dataset used for domain adaptation on which we evaluate on $\mathcal{D}'$

| Embedding Model | Baseline | | Standardized | | Whitened | | PCA | |
|---|---|---|---|---|---|---|---|---|
| | Acc | $I_A, I_B$ | Acc | $I_A, I_B$ | Acc | $I_A, I_B$ | Acc | $I_A, I_B$ |
| TelecomQuAD | | | | | | | | |
| bge_large | 66.87 | 9.24, 27.81 | 66.63 | 9.71, 97.23 | 65.11 | 9.41, 79.15 | 68.43 | 16.91, 95 |
| bge_large_pretrained | 62.64 | 6.34, 23.77 | 59.24 | 6.82, 96.26 | 63.17 | 6.78, 24.96 | 57.02 | 12.36, 92.75 |
| bge_large_finetuned | 81.61 | 11.45, 40.58 | 82.66 | 11.89, 97.54 | 82.03 | 11.87, 40.10 | 78.76 | 18.09, 97.99 |
| bge_large_pretrained_finetuned | 81.67 | 10.34, 45.27 | 80.48 | 10.78, 97.26 | 81.44 | 73.0, 88.0 | 77.46 | 15.54, 98.35 |
| llm_embedder | 70.06 | 10.83, 14.54 | 68.26 | 11.59, 96.83 | 69.66 | 11.59. 13.93 | 68.58 | 20.5, 96.71 |
| llm_embedder_pretrained | 57.12 | 5.42, 15.4 | 53.09 | 5.94, 95.77 | 56.56 | 47.0, 65.52 | 56.55 | 11.31, 95.77 |
| llm_embedder_finetuned | 81.58 | 13.94, 22.1 | 82.28 | 14.66, 97.34 | 81.52 | 14.63, 19.88 | 79.14 | 20.73, 97.78 |
| llm_embedder_pretrained_finetuned | 80.37 | 10.74, 25.01 | 81.2 | 11.25, 97.32 | 80.79 | 11.23,23.22 | 79.44 | 15.82, 98.11 |
| sciq | | | | | | | | |
| bge_large | 92.7 | 9.29, 82.99 | 93.2 | 9.58, 97.89 | 92.7 | 9.73, 82.77 | 89.4 | 30.41, 91.88 |
| bge_large_pretrained | 92.1 | 5.04, 85.08 | 92.6 | 5.29, 97.05 | 91.9 | 5.31, 85.13 | 86.4 | 31.95, 91.82 |
| bge_large_finetuned | 94.5 | 5.77, 73.93 | 94.8 | 6.21, 97.26 | 94.8 | 6.18, 73.94 | 89.8 | 24.63, 91.84 |
| bge_large_pretrained_finetuned | 95.5 | 6.83, 76.76 | 94.7 | 7.22, 97.51 | 95.5 | 7.21, 76.83 | 89.4 | 28.32, 91.82 |
| llm_embedder | 91.7 | 10.62, 67.87 | 93.0 | 10.96, 96.77 | 91.1 | 11.8, 66.38 | 86.9 | 29.65, 91.24 |
| llm_embedder_pretrained | 91.0 | 6.43, 73.75 | 91.3 | 6.78, 96.1 | 91.0 | 6.8, 72 | 83.7 | 25.18, 91.13 |
| llm_embedder_finetuned | 94.4 | 8.49, 65.9 | 93.9 | 9.06, 96.51 | 94.1 | 9.05, 65.25 | 86.8 | 28.81, 91.15 |
| llm_embedder_pretrained_finetuned | 94.0 | 10.08, 67.46 | 93.5 | 10.48, 96.92 | 93.5 | 10.57, 66.48 | 86.5 | 27.02, 91.13 |
| PubMedQA | | | | | | | | |
| bge_large | 94.0 | 11.09, 79.17 | 95.8 | 11.59, 97.83 | 93.7 | 11.48, 79 | 91.5 | 26.6, 92.78 |
| bge_large_pretrained | 84.3 | 4.96, 76.3 | 81.1 | 5.39, 95.8 | 81.1 | 84.07, 76.44 | 80.2 | 27.01, 92.76 |
| bge_large_finetuned | 98.5 | 8.25, 72.33 | 97.3 | 8.76, 97.18 | 98.2 | 8.76, 71.56 | 93.4 | 27.3, 92.77 |
| bge_large_pretrained_finetuned | 97.8 | 6.42, 72.42 | 96.8 | 6.92, 96.69 | 98.2 | 6.88, 72.52 | 94.0 | 27, 92.74 |
| llm_embedder | 95.4 | 14.93, 63.95 | 95.3 | 15.67, 96.75 | 95.3 | 15.6, 62.68 | 90.6 | 33.86, 90.33 |
| llm_embedder_pretrained | 91.9 | 6.41, 68.04 | 90.7 | 7.09, 94.19 | 91.4 | 7.08, 67.35 | 85.7 | 28.7, 90.19 |
| llm_embedder_finetuned | 97.5 | 12.81, 63.51 | 97.5 | 13.53, 96.42 | 97.9 | 13.53, 62.33 | 92.6 | 33.97, 90.39 |
| llm_embedder_pretrained_finetuned | 97.5 | 8.96, 63.96 | 97.1 | 9.77, 95.59 | 97.0 | 9.67, 62.5 | 91.8 | 32.72, 90.35 |

Table 5: Accuracy, $I_A$ and $I_B$ for embeddings under different transformations.

## 4.2 ISOTROPY SCORE ANALYSIS

Table 5 lists the retrieval accuracies for the telecom dataset $\mathcal{D}$, isotropic measures $I_A$ and $I_B$ of base and adapted models for various transformations (intended to increase isotropy scores and described in Section 2.5).

Correlation of $I_A$ and $I_B$ with accuracies across base, fine-tuned models with and without post-processing using transformations described in Section 2.5 is presented in Table 3. We see that, accuracy and both the isotropy scores are not correlated across datasets. Contrary to the conflicting claims in Jung et al. (2023) and Rudman & Eickhoff (2023), our experiments across domains establish that accuracy and isotropy scores are not correlated.

Combining these observations, we conclude that fine tuning improves the isotropy but isotropy cannot be attributed to retrieval accuracies. Our studies indicate that this may be the right resolution between the contradictions among studies by Jung et al. (2023) and Rudman & Eickhoff (2023) which we have discussed in Section 1.

### 4.3 EMBEDDING ANALYSIS POST DOMAIN ADAPTATION

As described in Section 2.6, Figure 2 shows the distribution of distances $\Delta_{\mathcal{M}}$ and $\Delta_{\mathcal{M}'}$ for both *bge_large* and *llm_embedder*, respectively. The plots show that post domain-adaptation, the embeddings move away from the domain-agnostic embeddings. For *llm_embedder*, the farthest points between $\mathcal{D}$ and $\mathcal{D}'$ on public models is closer than the closest ones post domain adaptation. For *bge_large*, it is clear that distance between embeddings increase post domain adaptation, but this is less pronounced on Sci-Q and PubMedQA datasets. This is one of the effects of domain-adaption and needs further study.

## 5 RECOMMENDATIONS AND CONCLUSIONS

### 5.1 RECOMMENDATIONS

In this work, we have done a series of experiments to establish the impact of domain adaptation for embedding models. Based on this, we provide a set of recommendations to a researcher/practitioner on best using our findings. We provide anonymized code[4] to perform the suggested steps, except domain adaptation, below

- Use a bootstrapped approach for obtaining accuracies as this will give not only point accuracies but also 95% confidence intervals.

- If possible, use domain adaptation - preferably pretraining followed by fine-tuning (PTFT)

- Identify thresholds for the similarity scores - this will lead to bootstrapped accuracy which is statistically same as the full dataset bootstrapped accuracy, while suppressing less relevant documents to end-users / downstream tasks

- We propose two new metrics COE and ROE. The observed correlations, across 3 datasets, of the COE with accuracy and the ROE with thresholds indicate that they are reliable measures for the generalisation of performance on unseen data of that domain.

- Our results establish the lack of correlation of accuracies to isotropy scores. We thus suggest that computing isotropy scores to interpret retrieval accuracies is unlikely to be beneficial.

### 5.2 CONCLUSION AND FUTURE WORK

We have reported mean bootstrapped retrieval accuracies along with confidence intervals for various SOTA embedding models with and without domain-adaptations. We observe that fine-tuning (with or without pre-training) improves both mean and CI of retrieval accuracies. However, pre-training followed by fine-tuning improves CI further. We propose a bootstrapped approach for choosing thresholds and observe that we can significantly reduce the number of retrieved sentences without any statistical deviation in retrieval performance. Our proposed cumulative distribution metrics, COE and ROE, to measure overlap between distributions of cosine similarities show strong correlations with retrieval performance and similarity thresholds respectively. We measure isotropy of embeddings using two independent SOTA isotropy metrics. We perform extensive evaluations on embeddings with and without isotropic transformations. We conclude that isotropy can be considered to be neither necessary nor sufficient from a retrieval accuracy perspective. Finally, we observe that with domain adaption, domain specific embedding show improved isotropy scores and move away from general domain embeddings. Our study establishes systematic methods of analysing embeddings in specialised domains. Our results hold across three different domain, which makes us believe that they hold for other specialised domains too. The current work considers QA task only. Future work may involve other tasks like summarization, or multimodal settings.

---

[4]https://anonymous.4open.science/r/embedingStudy-E3B5/

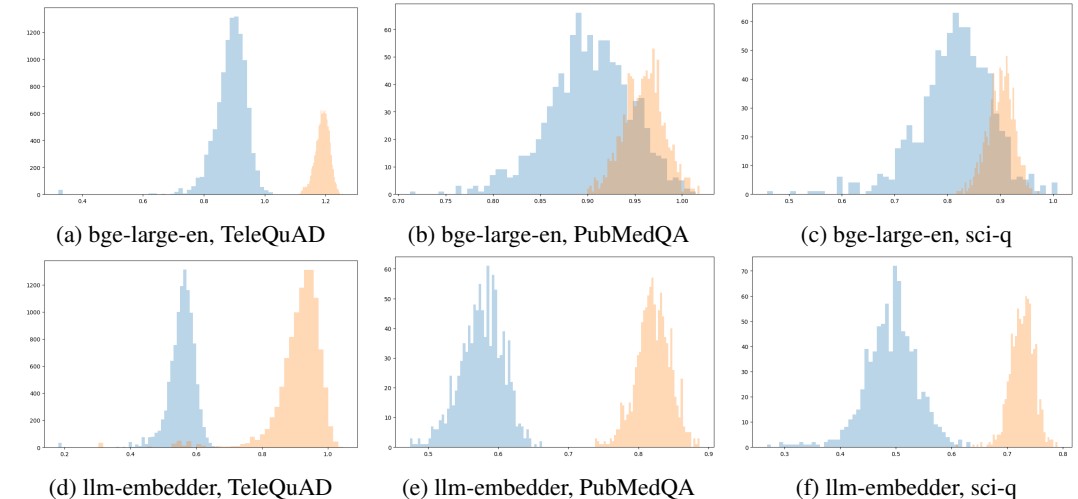

(a) bge-large-en, TeleQuAD    (b) bge-large-en, PubMedQA    (c) bge-large-en, sci-q

(d) llm-embedder, TeleQuAD    (e) llm-embedder, PubMedQA    (f) llm-embedder, sci-q

Figure 2: Distribution of distances for embedding models. Blue histogram represents $\Delta_{\mathcal{M}}$ and orange represents $\Delta_{\mathcal{M}'}$ using bge_large and llm_embedder models as $\mathcal{M}$. Refer Sec. 2.6 for definitions

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

# A APPENDIX

## A.1 DETAILED METRICS FOR VARIOUS DATASETS

| Embedding Model | Bootstrapped Acc (CI) | COE (CI) | ROE (CI) | $\tau, \psi$ | Acc @ $\tau$ |
|---|---|---|---|---|---|
| | Dataset: TeleQuAD, k=5 | | | | |
| gte | 72.55 (64.48, 81.0) | 87.66 (83.08, 92.35) | 0.31 (0.20, 1.02) | 0.28 (20) | 69.75 |
| bge_large | 66.87 (58.48, 75.52) | 87.98 (82.06, 94.70) | 4.81 (1.22, 18.04) | 0.5 (35) | 67.18 |
| bge_large_pretrained | 62.64 (53.0, 70.0) | 85.94 (79.92, 93.68) | 2.18 (0.61, 6.52) | 0.58 (25) | 61.36 |
| bge_large_finetuned | 81.61 (74.48, 88.52) | 91.98 (90.42, 94.09) | 0.22 (0.00, 0.82) | 0.43 (25) | 79.46 |
| bge_large_pretrained_finetuned | 81.67 (74.48, 87.52) | 91.06 (89.81, 93.07) | 0.23 (0.20, 0.31) | 0.4 (35) | 77.73 |
| llm_embedder | 70.06 (63.0, 77.52) | 87.26 (80.63, 96.13) | 5.77 (0.92, 26.71) | 0.78 (30) | 69.9 |
| llm_embedder_pretrained | 57.12 (47.48, 67.05) | 84.88 (79.71, 91.34) | 6.32 (1.94, 19.27) | 0.75 (30) | 52.53 |
| llm_embedder_finetuned | 81.58 (75.0, 88.52) | 90.73 (88.38, 95.11) | 0.10 (0.00, 0.82) | 0.56 (40) | 80.69 |
| llm_embedder_pretrained_finetuned | 80.37 (74.0, 86.52) | 90.74 (88.99, 94.70) | 0.21 (0.10, 0.41) | 0.53 (25) | 77.97 |
| ada_002 | 75.48 (68.47, 83.0) | 90.19 (85.02, 95.92) | 3.21 (1.33, 9.28) | 0.75 (25) | 75.3 |
| text-embedding-3-small | 69.91 (62.0, 79.0) | 87.59 (83.38, 95.01) | 1.90 (0.61, 8.05) | 0.31 (20) | 68.43 |
| text-embedding-3-large | 75.96 (69.0, 82.0) | 90.32 (86.03, 96.74) | 5.99 (2.14, 23.75) | 0.26 (20) | 73.72 |
| mpnet | 61.49 (54.0, 71.0) | 81.74 (75.54, 91.34) | 2.91 (0.82, 11.21) | 0.29 (45) | 59.78 |
| minilm | 67.26 (60.0, 75.0) | 83.25 (78.90, 90.72) | 0.77 (0.20, 3.47) | 0.27 (25) | 64.98 |
| | Dataset: sci-qa, k=5 | | | | |
| bge_large | 92.7 (87, 97) | 96.98 (95.85, 99.39) | 4.14 (2.2, 13.9) | 0.508 (85) | 92.8 |
| bge_large_pretrained | 92.08 (87, 96) | 95.48 (94.02, 97.8) | 1.99 (1.34, 4.39) | 0.5 (15) | 92.1 |
| bge_large_finetuned | 94.45 (90, 98) | 97.79 (96.95, 99.27) | 3.72 (2.07, 8.54) | 0.883 (85) | 94.8 |
| bge_large_pretrained_finetuned | 95.5 (91, 98) | 98.15 (97.2, 99.27) | 3.25 (1.71, 8.05) | 0.771 (85) | 95.4 |
| llm_embedder | 91.69 (87, 96) | 97.2 (95.49, 99.27) | 6.85 (2.93, 22.68) | 0.809 (85) | 91.8 |
| llm_embedder_pretrained | 91.03 (86, 96) | 94.42 (92.56, 98.17) | 2.21 (1.22, 5.61) | 0.682 (85) | 91.2 |
| llm_embedder_finetuned | 94.37 (90, 98) | 98.18 (97.32, 99.51) | 7.4 (3.17, 24.39) | 0.883 (85) | 94.7 |
| llm_embedder_pretrained_finetuned | 94 (90, 98) | 98.24 (97, 99.51) | 6.28 (2.32, 23.05) | 0.824 (85) | 93.9 |
| | Dataset: PubMedQA, k=5 | | | | |
| bge_large | 94 (89, 98) | 96.2 (95.1, 98.4) | 3.21 (1.3, 13.1) | 0.426 (5) | 93.8 |
| bge_large_pretrained | 84.3 (78, 90) | 93.8 (90.3, 99) | 7.52 (3.2, 23.8) | 0.492 (5) | 83.4 |
| bge_large_finetuned | 98.51 (95, 100) | 99.5 (99.3, 99.7) | 4.81 (2.8, 10.1) | 0.791 (85) | 98.5 |
| bge_large_pretrained_finetuned | 97.83 (96, 100) | 98.95 (98.4, 99.6) | 6.71 (3.2, 21) | 0.755 (45) | 97.7 |
| llm_embedder | 95.42 (91, 99) | 97.7 (97, 99.5) | 4.49 (1, 24.1) | 0.732 (5) | 95.8 |
| llm_embedder_pretrained | 91.87 (86, 96) | 96.94 (95.3, 99.2) | 4.1 (1.6, 18.7) | 0 (100) | 92.0 |
| llm_embedder_finetuned | 97.53 (94, 100) | 99.28 (98.8, 99.8) | 5.55 (2.5, 14.88) | 0.797 (30) | 97.5 |
| llm_embedder_pretrained_finetuned | 97.5 (95, 100) | 99 (98.6, 99.7) | 6.15 (3, 22.8) | 0.783 (30) | 97.2 |

Table 6: Different metrics for various datasets ($\mathcal{D}$), $K$=5

| Embedding Model | Baseline | | Standardized | | Whitened | | PCA | |
|---|---|---|---|---|---|---|---|---|
| | Acc | $I_A, I_B$ | Acc | $I_A, I_B$ | Acc | $I_A, I_B$ | Acc | $I_A, I_B$ |
| | | | TelecomQuAD | | | | | |
| gte | 72.55 | 11.33, 42.4 | 72.22 | 11.64, 96.75 | 71.98 | 11.64,43.14 | 70.85 | 21.34, 97.49 |
| bge_large | 66.87 | 9.24, 27.81 | 66.63 | 9.71, 97.23 | 65.11 | 9.41, 79.15 | 68.43 | 16.91, 95 |
| bge_large_pretrained | 62.64 | 6.34, 23.77 | 59.24 | 6.82, 96.26 | 63.17 | 6.78, 24.96 | 57.02 | 12.36, 92.75 |
| bge_large_finetuned | 81.61 | 11.45, 40.58 | 82.66 | 11.89, 97.54 | 82.03 | 11.87, 40.10 | 78.76 | 18.09, 97.99 |
| bge_large_pretrained_finetuned | 81.67 | 10.34, 45.27 | 80.48 | 10.78, 97.26 | 81.44 | 73.0,98.2 | 77.46 | 15.54, 98.35 |
| llm_embedder | 70.06 | 10.83, 14.54 | 68.26 | 11.59, 96.83 | 69.66 | 11.59. 13.93 | 68.58 | 20.5, 96.71 |
| llm_embedder_pretrained | 57.12 | 5.42, 15.4 | 53.09 | 5.94, 95.77 | 56.56 | 47.0, 65.52 | 56.55 | 11.31, 95.77 |
| llm_embedder_finetuned | 81.58 | 13.94, 22.1 | 82.28 | 14.66, 97.34 | 81.52 | 14.63, 19.88 | 79.14 | 20.73, 97.78 |
| llm_embedder_pretrained_finetuned | 80.37 | 10.74, 25.01 | 81.2 | 11.25, 97.32 | 80.79 | 11.23,23.22 | 79.44 | 15.82, 98.11 |
| ada_002 | 75.48 | 6.64, 25.52 | 68.83 | 7.08, 97.09 | 75.46 | 7.06.26.18 | 69.31 | 15.67, 93.8 |
| text-embedding-3-small | 69.91 | 6.33, 45.86 | 67.18 | 6.79, 93.14 | 69.52 | 6.72, 45.88 | 66.26 | 14.46, 89.17 |
| text-embedding-3-large | 75.96 | 3.64, 63.78 | 71.47 | 4.11, 94.38 | 74.84 | 4.08, 64.33 | 70.94 | 11.83, 89.12 |
| mpnet | 61.49 | 9.51, 36.41 | 57.62 | 10.25, 96.66 | 59.99 | 10.22, 27.68 | 56.39 | 16.35, 93.72 |
| minilm | 67.26 | 19.43, 26.59 | 63.99 | 20.54, 95.56 | 66.19 | 20.37,26.36 | 62.22 | 25.67, 95.32 |
| | | | sciq | | | | | |
| bge_large | 92.7 | 9.29, 82.99 | 93.2 | 9.58, 97.89 | 92.7 | 9.73, 82.77 | 89.4 | 30.41, 91.88 |
| bge_large_pretrained | 92.1 | 5.04, 85.08 | 92.6 | 5.29, 97.05 | 91.9 | 5.31, 85.13 | 86.4 | 31.95, 91.82 |
| bge_large_finetuned | 94.5 | 5.77, 73.93 | 94.8 | 6.21, 97.26 | 94.8 | 6.18, 73.94 | 89.8 | 24.63, 91.84 |
| bge_large_pretrained_finetuned | 95.5 | 6.83, 76.76 | 94.7 | 7.22, 97.51 | 95.5 | 7.21, 76.83 | 89.4 | 28.32, 91.82 |
| llm_embedder | 91.7 | 10.62, 67.87 | 93.0 | 10.96, 96.77 | 91.1 | 11.8, 66.38 | 86.9 | 29.65, 91.24 |
| llm_embedder_pretrained | 91.0 | 6.43, 73.75 | 91.3 | 6.78, 96.1 | 91.0 | 6.8, 72 | 83.7 | 25.18, 91.13 |
| llm_embedder_finetuned | 94.4 | 8.49, 65.9 | 93.9 | 9.06, 96.51 | 94.1 | 9.05, 65.25 | 86.8 | 28.81, 91.15 |
| llm_embedder_pretrained_finetuned | 94.0 | 10.08, 67.46 | 93.5 | 10.48, 96.92 | 93.5 | 10.57, 66.48 | 86.5 | 27.02, 91.13 |
| | | | PubMedQA | | | | | |
| bge_large | 94.0 | 11.09, 79.17 | 95.8 | 11.59, 97.83 | 93.7 | 11.48, 79 | 91.5 | 26.6, 92.78 |
| bge_large_pretrained | 84.3 | 4.96, 76.3 | 81.1 | 5.39, 95.8 | 81.1 | 84.07, 76.44 | 80.2 | 27.01, 92.76 |
| bge_large_finetuned | 98.5 | 8.25, 72.33 | 97.3 | 8.76, 97.18 | 98.2 | 8.76, 71.56 | 93.4 | 27.3, 92.77 |
| bge_large_pretrained_finetuned | 97.8 | 6.42, 72.42 | 96.8 | 6.92, 96.69 | 98.2 | 6.88, 72.52 | 94.0 | 27, 92.74 |
| llm_embedder | 95.4 | 14.93, 63.95 | 95.3 | 15.67, 96.75 | 95.3 | 15.6, 62.68 | 90.6 | 33.86, 90.33 |
| llm_embedder_pretrained | 91.9 | 6.41, 68.04 | 90.7 | 7.09, 94.19 | 91.4 | 7.08, 67.35 | 85.7 | 28.7, 90.19 |
| llm_embedder_finetuned | 97.5 | 12.81, 63.51 | 97.5 | 13.53, 96.42 | 97.9 | 13.53, 62.33 | 92.6 | 33.97, 90.39 |
| llm_embedder_pretrained_finetuned | 97.5 | 8.96, 63.96 | 97.1 | 9.77, 95.59 | 97.0 | 9.67, 62.5 | 91.8 | 32.72, 90.35 |

Table 7: Accuracy, $I_A$ and $I_B$ for embeddings under transformations, detailed results.

## A.2 SAMPLE QUESTIONS FROM TELECOMQUAD

| Question | Answer |
|---|---|
| What does AoA stand for? | AoA stands for Angle of Arrival. |
| Who specifies the chargeable events? | Chargeable events are specified by middle tier TS. |
| What architecture is defined in 3GPP TS 32.240 V15.5.0? | Single IMSI Architecture is defined in 3GPP TS 32.240 V15.5.0. |

Table 8: Sample questions and their correct answers - based on 3GPP Release 17.

## A.3 ISOTROPY SCORES

We provide brief details of isotropy measures used in this work. According to Mu et al. (2017), isotropy $I_a(v)$ can be defined by (3).

$$I_a(v) = \frac{\min_{|c|=1} Z(c)}{\max_{|c|=1} Z(c)}, \tag{3}$$

where $Z(c)$ is a partition function as given by (4)

$$Z(c) = \sum_{\forall v} exp(c^T v), \tag{4}$$

such that $Z(c)$ is constant with any unit function.

If $V$ is the matrix stacked by all embedding vectors, $1_{|V|}$ be the vectors with all entries equal to one, then the second order approximation of isotropy is given by (5).

$$I_A(\{v\}) = \frac{|V| - ||I_{|V|}^T V|| + 0.5\sigma_{min}^2}{|V| + ||I_{|V|}^T V|| + 0.5\sigma_{max}^2}, \tag{5}$$

where $\sigma_{min}$ and $\sigma_{max}$ correspond to smallest and largest singular value of $V$, respectively.

Further, Rudman et al. (2021) define isotropy as given by (6).

$$I_B(v) = \frac{(n - \delta(v)^2(n - \sqrt{n}))^2 - n}{n(n-1)}, \tag{6}$$

where for any embedding $v \in \mathbb{R}^n$, $\delta(v) = \frac{||\Sigma_D - 1||}{\sqrt{2(n - \sqrt{n})}}$, $1 = (1, 1, ..., 1)^T \in \mathbb{R}^n$, and $\Sigma_D \in \mathbb{R}^n$ is the normalized diagonal of covariance matrix of $PCA(V)$, which is the set of embeddings transformed by the first $n$ principal components.

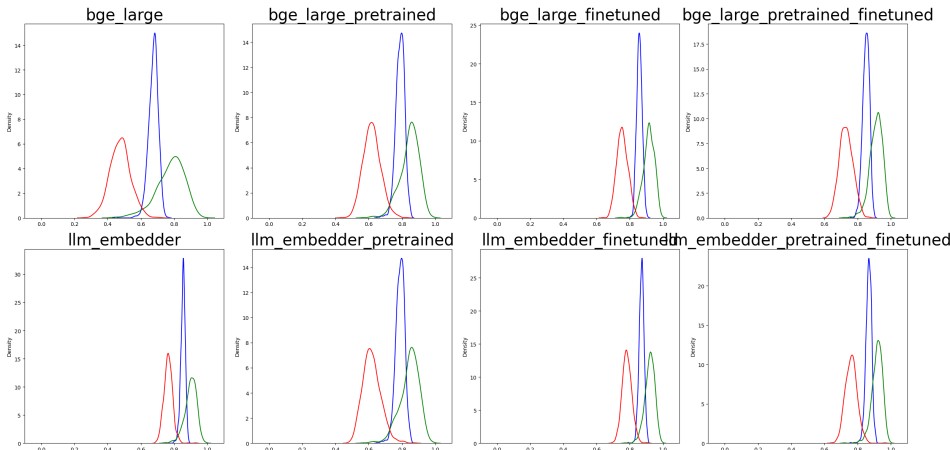

Figure 3: Density plots for health dataset. Red, green and blue indicate distribution of $S_{rand}$, $S_{corr}$ and $S_{topK}$ respectively. Refer Sec. 2.3 for definitions

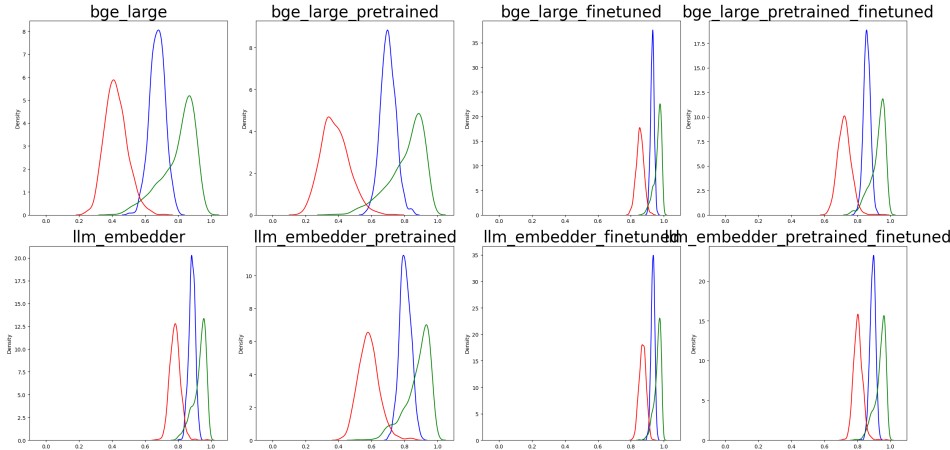

Figure 4: Density plots for science dataset. Red, green and blue indicate distribution of $S_{rand}$, $S_{corr}$ and $S_{topK}$ respectively

## A.4 DISTRIBUTION PLOTS

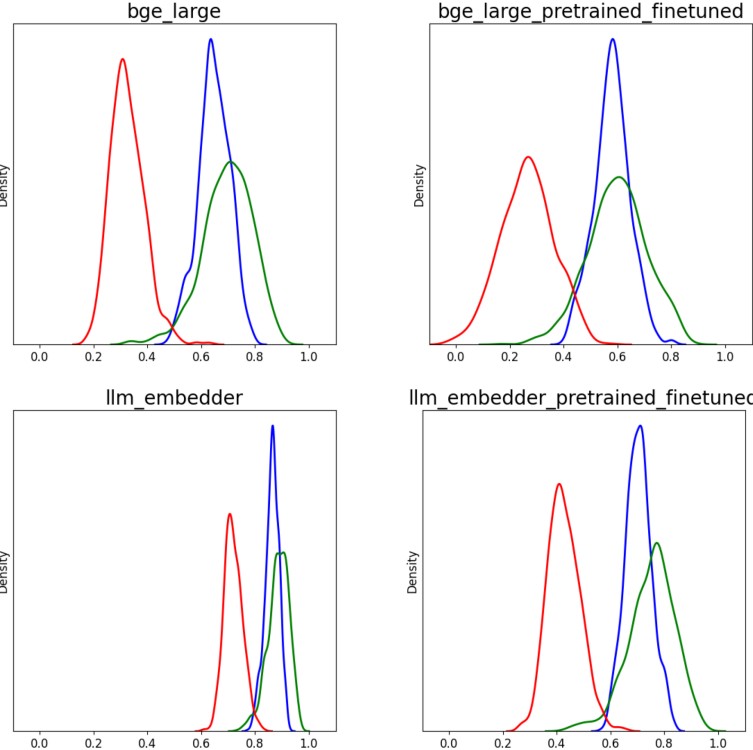

Figure 5: Density plots for SQuAD dataset $\mathcal{D}'$ - the PTFT model is domain adapted using telecom data. Red, green and blue indicate distribution of $S_{rand}$, $S_{corr}$ and $S_{topK}$ respectively

## A.5 NDCG SCORES

Table 9 shows Normalized Discounted Cummulative Gain (NDCG) Wang et al. (2013) scores for various datasets for base, PT, FT and PT-FT variants of embedding models. The columns indicate the NDCG score, lower, upper bound and width of confidence interval, as well as accuracy and NDCG on the full dataset.

We observe that the bootstrapped accuracy numbers are comparable to those on the full dataset, this validates our approach. Further, if we consider the NDCG scores, they improve with domain adaptation. NDCG scores are relatively lower because we have only one correct answer per question. However, the improvement of NDCG scores show that not only do we get tighter confidence intervals on domain adaptation, our correct answers are retrieved with better ranks.

| Model | Variant | TeleQuAD | | | | | |
|---|---|---|---|---|---|---|---|
| | | NDCG | NDCG-LOW | NDCG-HIGH | CI WIDTH | Full Data Acc | Full Data NDCG |
| bge-large | Base | 29.6 | 29.4 | 30.0 | 0.6 | 66.0 | 29.9 |
| | PT | 27.2 | 27.0 | 27.4 | 0.4 | 63.1 | 27.5 |
| | FT | 34.2 | 33.2 | 34.3 | 1.2 | 82.0 | 34.2 |
| | PT-FT | 34.9 | 34.8 | 35.4 | 0.5 | 81.5 | 34.9 |
| LLM-embedder | Base | 29.2 | 27.9 | 29.5 | 1.6 | 69.2 | 29.3 |
| | PT | 25.2 | 24.6 | 25.3 | 0.8 | 57.0 | 25.2 |
| | FT | 34.3 | 33.8 | 34.4 | 0.6 | 81.8 | 34.4 |
| | PT-FT | 33.7 | 33.3 | 33.8 | 0.5 | 80.8 | 33.8 |
| | | PubMedQA | | | | | |
| | | NDCG | NDCG-LOW | NDCG-HIGH | CI WIDTH | Full Data Acc | Full Data NDCG |
| bge-large | Base | 37.2 | 37.2 | 37.3 | 0.1 | 94.0 | 37.2 |
| | PT | 35.1 | 34.9 | 35.2 | 0.4 | 84.3 | 35.0 |
| | FT | 38.5 | 38.3 | 38.5 | 0.3 | 97.8 | 38.5 |
| | PT-FT | 38.9 | 38.9 | 38.9 | 0.1 | 98.5 | 38.9 |
| LLM-embedder | Base | 37.5 | 37.4 | 37.7 | 0.3 | 95.4 | 37.5 |
| | PT | 37.0 | 36.8 | 37.2 | 0.4 | 91.8 | 37.0 |
| | FT | 38.1 | 38.1 | 38.2 | 0.1 | 97.6 | 38.1 |
| | PT-FT | 38.1 | 37.9 | 38.2 | 0.3 | 97.3 | 38.1 |
| | | Sci-QA | | | | | |
| | | NDCG | NDCG-LOW | NDCG-HIGH | CI WIDTH | Full Data Acc | Full Data NDCG |
| bge-large | Base | 36.8 | 36.8 | 36.9 | 0.1 | 92.8 | 36.9 |
| | PT | 36.7 | 36.6 | 36.8 | 0.2 | 92.1 | 36.7 |
| | FT | 37.4 | 37.2 | 38.5 | 1.4 | 94.5 | 37.5 |
| | PT-FT | 37.8 | 37.7 | 37.9 | 0.2 | 95.4 | 37.9 |
| LLM-embedder | Base | 36.9 | 36.8 | 37.2 | 0.4 | 91.8 | 36.8 |
| | PT | 36.3 | 36.1 | 36.4 | 0.3 | 91.0 | 36.3 |
| | FT | 37.7 | 37.6 | 37.9 | 0.3 | 94.0 | 37.6 |
| | PT-FT | 37.7 | 37.5 | 37.7 | 0.1 | 94.0 | 37.7 |

Table 9: NDCG Scores for various datasets.