# OpenReview forum: "Towards Understanding Domain Adapted Sentence Embeddings for Document Retrieval"
_ICLR.cc/2025/Conference — Submitted to ICLR 2025_

### Official Review · Reviewer_Ypbg · 2024-10-29

**Soundness:** 2
**Presentation:** 1
**Contribution:** 1
**Rating:** 3
**Confidence:** 4

**Summary:**

This paper investigates domain-adapted sentence embeddings for document retrieval. It proposes a systematic method to obtain thresholds for similarity scores for different embeddings, and proves that fine-tuning and pretraining-then-fine-tuning help improve retrieval accuracy through extensive experiments.

**Strengths:**

- This paper proposes a systematic method to introduce thresholds to improve document retrieval performance.
- Extensive experiments are conducted to demonstrate that fine-tuning and pretraining-then-fine-tuning help improve retrieval accuracy.

**Weaknesses:**

- Lack of novelty: Much existing work has shown that fine-tuning or continued pretraining helps to improve domain performance [1]. The isotropy problem has been well studied in [2][3] and many other contrastive learning-based sentence embeddings.
- There are too many research questions in this paper. It looks like the RQ2 and RQ3 don't have a strong connection to the research topic, i.e., domain-adapted sentence embeddings.
- Poor organization. The current introductory section functions as a literature review. It is suggested to clarify the research problem, research gap and contribution in the introduction.


**Reference:**

[1] Gururangan, S., Marasović, A., Swayamdipta, S., Lo, K., Beltagy, I., Downey, D., & Smith, N. A. (2020). Don't stop pretraining: Adapt language models to domains and tasks. arXiv preprint arXiv:2004.10964.

[2] Kawin Ethayarajh. 2019. How contextual are contextualized word representations? Comparing the geometry of BERT, ELMo, and GPT-2 embeddings. In Proceedings of the 2019 Conference on Empirical Methods in Natural Language Processing and the 9th International Joint Conference on Natural Language Processing (EMNLP-IJCNLP), pages 55–65, Hong Kong, China. Association for Computational Linguistics.

[3] Tianyu Gao, Xingcheng Yao, and Danqi Chen. 2021. Simcse: Simple contrastive learning of sentence embeddings. In Proceedings of the 2021 Conference on Empirical Methods in Natural Language Processing, pages 6894–6910. Association for Computational Linguistics.

**Questions:**

- It is unclear why the CI is set to 95%. Is there any supporting evidence?
- The ndcg is widely used to evaluate information retrieval performance. Why not use it?

---

> ### Author Response · Authors · 2024-11-24
>
> Following are our responses to the __weaknesses__ indicated by the reviewer:
>
> 1. Lack of novelty: Much existing work has shown that fine-tuning or continued pretraining helps to improve domain performance [1]. The isotropy problem has been well studied in [2][3] and many other contrastive learning-based sentence embeddings.
>
> __Whilst we agree that much work has shown the improvement with pretraining and fine tuning, there is no work, to the best of our knowledge, which looks at the effect of the same on confidence intervals. Also on isotropy, The results from the literature are conflicting - as detailed in the introduction with retrieval accuracies being shown to improve with isotropy [(Jung et al., 2023)](https://dl.acm.org/doi/abs/10.1007/978-3-031-33380-4_10) as well as anisotropy [(Rudman & Eickhoff, 2023)](https://iclr.cc/media/iclr-2024/Slides/18254_RTn9zZz.pdf) – we aim to resolve this via our work.__
>
> 2. There are too many research questions in this paper. It looks like the RQ2 and RQ3 don't have a strong connection to the research topic, i.e., domain-adapted sentence embeddings.
>
> __The research questions are looking at the effect of domain adaptation on sentence embeddings from three perspectives – confidence intervals on retrieval accuracies, our new metrics CoE and RoE and finally isotropy scores.__
>
> 3. Poor organization. The current introductory section functions as a literature review. It is suggested to clarify the research problem, research gap and contribution in the introduction.
>
> __We have revised the manuscript to address the concerns raised. The following is the structure :__
>
> 1. __Section 1 introduces the problem statement and describes prior work.__
>
>    1.1 __We have introduced a paragraph to clearly articulate the gaps in current practice, which leads to the research questions (Sec 1.1).__
>
> 2. __Following this, methodological details are provided in Section 2. Datasets are described in Section 3, experimental results are provided in Section 4.__
>
> 3. __We conclude in Section 5, including a Section (5.1), providing recommendations for both researchers and practitioners that can be leveraged for multiple use-cases.__
>
>
> _______________________________________________________________
>
> Following are our responses to the __questions__ indicated by the reviewer:
>
> •	It is unclear why the CI is set to 95%. Is there any supporting evidence?
>
> __95% CI is commonly used in statistical research and we have continued our research based on the same.__
>
> •	The ndcg is widely used to evaluate information retrieval performance. Why not use it?
>
> __Our experiments are on data with only one correct answer and we are looking at whether the correct is retrieved or not. In this context, NDCG is of lesser significance than accuracy. However, as you have raised a valid point on evaluating NDCG scores, we have performed additional experiments for the same. Results are reported in Appendix A.5 of the revised manuscript.__
>
> __We observe that the bootstrapped accuracy numbers are comparable to those on the full dataset, this validates our approach. Further, if we consider the NDCG scores, they improve with domain adaptation. NDCG scores are relatively lower because we have only one correct answer per question. However, the improvement of NDCG scores show that not only do we get tighter confidence intervals on domain adaptation, our correct answers are retrieved with better ranks.__

---

> > ### Comment · Reviewer_Ypbg · 2024-11-25
> > **Reply to authors**
> >
> > Thank you for your responses. However, my concerns have not been addressed. I prefer to keep the score unchanged.

---

### Official Review · Reviewer_PhJW · 2024-11-02

**Soundness:** 2
**Presentation:** 1
**Contribution:** 1
**Rating:** 3
**Confidence:** 4

**Summary:**

The paper presents an empirical study of domain adapted embedding models for QA. The authors evaluate a range of existing models and their domain-adapted variants on point accuracy and confidence intervals of retrieval tasks. The main contributions of this study are using the following aspects to provide a recommendation to select retrieval thresholds for domain adapted models:

1. A methodology for determining appropriate similarity score thresholds for different embedding models.
2. Introduction of new metrics that capture the distributional overlap between top-ranked, correct, and randomly selected document with respect to the input question.
3. Analysis of the relationship between embedding isotropy (uniformity of vector lengths) and retrieval accuracy.
4. Observations that domain-adapted embeddings exhibit little overlap with their domain-agnostic counterparts.

**Strengths:**

The authors define a relevant problem for RAG applications and show that fine-tuning can help to not only improve the retrieval performance but also increase the confidence intervals tightens. Thus, supporting that domain-adaptation is likely to also reduce the expected performance variance in that domain.

**Weaknesses:**

While the problem addressed is relevant and some of the findings hold intrinsic interest, the paper also presents several notable limitations that should be addressed.

First, the confidence interval (CI) estimation techniques employed are direct applications of common methodologies to measure CI for a given metric (in this case accuracy).

Second, the paper does not provide a compelling justification for the proposed assessment to select the retrieval threshold, nor does it rigorously compare their performance to simpler approaches such as hyper-parameter cross-validation which is relatively inexpensive assuming the similarity matrix is computed only once. The added value of the new techniques is therefore unclear from the current presentation.

Third, while the paper cites prior studies on the concept of isotropy, there is insufficient explanation of how the experimental setup and findings in this work relate to or build upon those earlier investigations. The connection to the broader academic context is not well established.

Finally, the overall narrative lacks cohesion, as the sections are not clearly tied together, and the suggested methodology for assessing retrieval thresholds is not systematically contrasted against relevant baseline approaches. This makes it difficult for the reader to fully grasp the unique contributions of the work.

**Questions:**

The paper could benefit from a more cohesive narrative and clearer articulation of the benefits of the proposed assessment approach. While the common theme of domain adaptation links the various experiments, it is unclear what specific advantages the presented methodology offers.

To strengthen the paper, I would suggest addressing the following points:

Clearly define the end goal of the work. For example, if the aim is to provide guidance on setting appropriate retrieval thresholds, what is the current standard practice in this area? How does the proposed method improve upon existing approaches?

Explicitly highlight the key advantages of the new assessment technique. What performance gains or other benefits does it provide compared to baseline methods?

Structure the paper to guide the reader seamlessly from the problem statement, to the methodological details, and finally to a clear conclusion about the merits and recommended use cases of the new assessment strategy.

---

> ### Author Response · Authors · 2024-11-24
>
> __(1/2)__
>
> Following are our responses to the __weaknesses__ indicated by the reviewer:
>
> 1. First, the confidence interval (CI) estimation techniques employed are direct applications of common methodologies to measure CI for a given metric (in this case accuracy).
>
> __To the best of our knowledge, this has not been established in any literature either empirically or theoretically and so we consider this to be an important contribution and the effect of domain adaptation on CIs is of importance to both researchers and practitioners.__
>
> 2. Second, the paper does not provide a compelling justification for the proposed assessment to select the retrieval threshold, nor does it rigorously compare their performance to simpler approaches such as hyper-parameter cross-validation which is relatively inexpensive assuming the similarity matrix is computed only once. The added value of the new techniques is therefore unclear from the current presentation.
>
> __In a retrieval task for QA, especially with a single relevant document (answer), addition of any threshold will lead to poor performance so long as the threshold is higher than the minimum correct threshold. To the best of our knowledge, we have not found prior work which has provided a systematic approach to choose the best threshold. In practice, such thresholds are often chosen by inspection of similarity scores. Our approach of bootstrapping provides the ability to perform tests for statistical significance on the results, and we choose the maximum threshold such that our results are not statistically worse off.__
>
> __Hyper-parameter tuning / cross-validation is very efficient for classification problems where there is a trade-off with the choice of threshold. Unfortunately, as accuracy scores reduce, this trade-off is less relevant for retrieval tasks with one relevant document.__
>
> 3. Third, while the paper cites prior studies on the concept of isotropy, there is insufficient explanation of how the experimental setup and findings in this work relate to or build upon those earlier investigations. The connection to the broader academic context is not well established.
>
> __The results from the literature are conflicting - as detailed in the introduction with retrieval accuracies being shown to improve with isotropy [(Jung et al., 2023)](https://dl.acm.org/doi/abs/10.1007/978-3-031-33380-4_10) as well as anisotropy [(Rudman & Eickhoff, 2023)](https://iclr.cc/media/iclr-2024/Slides/18254_RTn9zZz.pdf) – we aim to resolve this via our work.__
>
>
> 4. Finally, the overall narrative lacks cohesion, as the sections are not clearly tied together, and the suggested methodology for assessing retrieval thresholds is not systematically contrasted against relevant baseline approaches. This makes it difficult for the reader to fully grasp the unique contributions of the work.
>
> __We have revised the manuscript to address the concerns raised. The following is the structure :__
>
> 1. __Section 1 introduces the problem statement and describes prior work.__
>
>    1.1. __We have introduced a paragraph to clearly articulate the gaps in current practice, which leads to the research questions (Sec 1.1).__
>
> 2. __Following this, methodological details are provided in Section 2. Datasets are described in Section 3, experimental results are provided in Section 4.__
>
> 3. __We conclude in Section 5, including a Section (5.1), providing recommendations for both researchers and practitioners that can be leveraged for multiple use-cases.__

---

> > ### Comment · Reviewer_PhJW · 2024-11-24
> >
> > Thank you for addressing my comments in the weaknesses section. However, the current version of the paper does not yet fully address my concerns. For now, I will maintain my score but will review the updated version you mentioned once it becomes available.

---

> > ### Author Response · Authors · 2024-11-28
> >
> > __(2/2)__
> >
> > Following are our responses to the __questions__ indicated by the reviewer:
> >
> > Clearly define the end goal of the work. For example, if the aim is to provide guidance on setting appropriate retrieval thresholds, what is the current standard practice in this area? How does the proposed method improve upon existing approaches?
> >
> > __End Goal: The end goal of our work is to provide a systematic method to determine retrieval scores and choose thresholds in a way that statistical confidence intervals and tests of significance are valid. We have also provided a set of recommendations for this in Section 5.1, which would be useful for both researchers and practitioners. We have evaluated our approach on datasets from multiple domains to demonstrate robustness and validity of the proposed approach.__
> >
> > __Standard Practice: In practice, such thresholds are often chosen by inspection of similarity scores.__
> >
> > __Proposed Improvement: In a retrieval task for QA, especially with a single relevant document/answer, addition of any threshold will lead to poor performance so long as the threshold is higher than the minimum correct threshold. To the best of our knowledge, we have not found prior work which has provided a systematic approach to choose the best threshold. In practice, such thresholds are often chosen by inspection of similarity scores. Our approach of bootstrapping provides the ability to perform tests for statistical significance on the results, and we choose the maximum threshold such that our results are not statistically worse off.__
> >
> > Explicitly highlight the key advantages of the new assessment technique. What performance gains or other benefits does it provide compared to baseline methods?
> >
> > __The first advantage that our method identifies is the ability to get confidence intervals and perform statistical tests on retrieval scores.__
> >
> > __This also allows us to establish a systematic method for identifying thresholds without affecting statistical significance of scores. Our new metrics (COE and ROE) provide estimates of the overlap between distributions of correct and random similarity scores with the top-k retrieved scores. We establish correlations of these with accuracy and thresholds, respectively.__
> >
> > __Finally, to understand the effect of domain adaptation, we analyze how the domain-specific embeddings move apart from general domain embeddings post domain adaptation.__
> >
> > __We perform experiments on three different domains and find that our observations hold true across them.__
> >
> > Structure the paper to guide the reader seamlessly from the problem statement, to the methodological details, and finally to a clear conclusion about the merits and recommended use cases of the new assessment strategy.
> >
> > __We have revised the manuscript to address the concerns raised. The following is the structure :__
> >
> > 1. __Section 1 introduces the problem statement and describes prior work.__
> >
> >    1.1 __We have introduced a paragraph to clearly articulate the gaps in current practice, which leads to the research questions (Sec 1.1).__
> >
> > 2. __Following this, methodological details are provided in Section 2. Datasets are described in Section 3, experimental results are provided in Section 4.__
> >
> > 3. __We conclude in Section 5, including a Section (5.1), providing recommendations for both researchers and practitioners that can be leveraged for multiple use-cases.__

---

### Official Review · Reviewer_8dAR · 2024-11-04

**Soundness:** 3
**Presentation:** 3
**Contribution:** 1
**Rating:** 3
**Confidence:** 4

**Summary:**

This paper investigates different aspects of sentence encoders for domain-specific question-answer retrieval. It aims to address the limitations of cosine similarity in capturing true semantic similarity, especially for frequent words with homonyms or dependence on regularization. To improve retrieval evaluation, the authors use confidence intervals (CIs) over bootstrapped samples for models before and after pre-training and fine-tuning on specific domains. They introduce COE and ROE metrics to measure distributional overlaps between top-K, correct, and random document similarities with the query, showing these metrics correlate with accuracy and threshold values. The paper visualizes how domain adaptation shifts embeddings away from domain-agnostic spaces. It also shows isotropy scores have limited correlation with retrieval performance.

**Strengths:**

1. The paper proposes novel metrics (CI, COE and ROE) over batched samples to evaluate embeddings for technical domains.
2. The methodology and metrics introduced are well-explained.

**Weaknesses:**

1. The paper lacks specifics on the pre-training corpus for each domain. If models are pre-trained solely on training sets, this limits the utility of pre-training, affecting the reliability of conclusions. For instance, in the health domain, pre-training on a large corpus like PubMed abstracts is common to ensure domain knowledge.
2. Comparisons are primarily with pre-trained models from BAAI and OpenAI; adding more state-of-the-art domain-specific baselines could strengthen the evaluation.
3. While the work presents interesting findings, the novelty is limited. Observations like tighter CIs with fine-tuning are expected since task-specific fine-tuning generally increases confidence for a specific task while potentially reducing generalizability.

**Questions:**

1. How does the test or dev set accuracy compare to bootstrapped accuracy? could you compare it with existing QA model results as a relaxed version of exact answer retrieval?

---

> ### Author Response · Authors · 2024-11-24
>
> Following are our responses to the __weaknesses__ indicated by the reviewer:
>
> 1. The paper lacks specifics on the pre-training corpus for each domain. If models are pre-trained solely on training sets, this limits the utility of pre-training, affecting the reliability of conclusions. For instance, in the health domain, pre-training on a large corpus like PubMed abstracts is common to ensure domain knowledge.
>
> __Whilst it is true that the maximum benefit of pretraining, we use a large corpus – we do our experiments on the same corpus as the fine tuning data because in many cases researchers and practitioners would be limited due to data and compute restrictions. We show improvements of pretrained finetuned embedding models over only finetuned models despite this limitation.__
>
> 2. Comparisons are primarily with pre-trained models from BAAI and OpenAI; adding more state-of-the-art domain-specific baselines could strengthen the evaluation.
>
> __The [BAAI models](https://github.com/netease-youdao/BCEmbedding) were among the top models on the MTEB scores.__
>
> 3. While the work presents interesting findings, the novelty is limited. Observations like tighter CIs with fine-tuning are expected since task-specific fine-tuning generally increases confidence for a specific task while potentially reducing generalizability.
>
> __Tighter CIs on fine tuning has not been reported in the literature prior to our knowledge and we are not aware of any theoretical work which establishes this fact as expected also.__
>
> Questions:
>
> 1. How does the test or dev set accuracy compare to bootstrapped accuracy? could you compare it with existing QA model results as a relaxed version of exact answer retrieval?
>
> __We have added these results in Appendix A.5 (see column _Full Acc Data_) to compare test set accuracy with bootstrapped accuracy. Additions are indicate in blue in the revised paper.__

---

> > ### Comment · Reviewer_8dAR · 2024-12-03
> >
> > 1. Thanks for clarifying the pre-training setup. However, since the goal is to evaluate the effect of pre-training, fine-tuning, and their combination, pre-training should be performed properly on a broader domain corpus rather than being restricted to downstream task data. Limiting pre-training in this way may lead to inaccurate conclusions.
> >
> > 3. While performance on tighter confidence intervals was not explicitly reported, this outcome aligns with general expectations about the effects of fine-tuning, as noted in my earlier comments.
> >
> > Thank you for addressing the weaknesses I raised. However, my concerns remain partially addressed. I will maintain my current score and reassess after the additional experimental results in your reply is available.

---

### Official Review · Reviewer_URNk · 2024-11-06

**Soundness:** 3
**Presentation:** 3
**Contribution:** 3
**Rating:** 8
**Confidence:** 3

**Summary:**

This paper reported mean bootstrapped retrieval accuracies along with confidence intervals for various SOTA embedding models with and without domain-adaptations. The results show that domain specific embedding show improved isotropy scores and move away from general domain embeddings. Overall, this is a solid study with reasonable technical contributions.

**Strengths:**

1. The paper is generally understandable and clearly explains the technical parts to a certain extent.
2. The figures and charts in the manuscript are exceptionally clear and well-presented.

**Weaknesses:**

1. This paper does not provide sufficient details on the ISOTROPY SCORES.
2. The paper does not sufficiently clarify the motivation behind its approach, especially regarding the "COMPUTATION OF THRESHOLDS". It lacks a detailed discussion on why existing methods struggle with this issue and how this paper effectively addresses it. it requires more illustrations and provements about this to help understanding the issue of existing methods and the motivation of the proposed new method in the paper.

**Questions:**

1. This paper does not provide sufficient details on the ISOTROPY SCORES.
2. The paper does not sufficiently clarify the motivation behind its approach, especially regarding the "COMPUTATION OF THRESHOLDS". It lacks a detailed discussion on why existing methods struggle with this issue and how this paper effectively addresses it. it requires more illustrations and provements about this to help understanding the issue of existing methods and the motivation of the proposed new method in the paper.

---

> ### Author Response · Authors · 2024-11-24
>
> We intend to make the following revisions to address questions raised:
>
> 1. This paper does not provide sufficient details on the ISOTROPY SCORES.
>
> __We have included details on computation of isotropy scores in Appendix A.3. The additions are indicated in blue in the revised paper.__
>
> 2. The paper does not sufficiently clarify the motivation behind its approach, especially regarding the "COMPUTATION OF THRESHOLDS". It lacks a detailed discussion on why existing methods struggle with this issue and how this paper effectively addresses it. it requires more illustrations and provements about this to help understanding the issue of existing methods and the motivation of the proposed new method in the paper.
>
> __Introduction of a threshold necessarily means that there will be a reduction in accuracy so long as the threshold is lower than the minimum cosine similarity for relevant (correct) documents to any question in the dataset. However, our approach finds the maximum possible threshold such that the bootstrapped accuracy post thresholding is not statistically different from the bootstrapped accuracy on the whole dataset. Our approach of using a bootstrapped estimate allows us to get statistical significance of results and thus our results are necessarily not worse than that without a threshold.__

---

### Meta-Review · Area_Chair_PqPh · 2024-12-21

**Metareview:**

This paper introduces a novel evaluation setup for the retrieval problem that includes both retrieval accuracy and confidence intervals. The paper measures these metrics for a diverse range of embedding methods, including both domain-agnostic ones and their domain-specific fine-tuned counterparts. The findings demonstrate that domain-specific fine-tuning not only improves retrieval accuracy but also results in tighter confidence intervals. Additionally, the paper highlights that domain-specific models achieve improved isotropy scores and differentiate embeddings from their domain-agnostic counterparts.

Strengths
- Domain adaptation, the focus of this paper, is an important problem (PhJW).
- The paper introduces a series of new metrics that are valuable (8dAR, Ypbg).
- Figures and graphs are clear and effective (URNk).

Weaknesses
- Insufficient details about certain metrics and data (URNk, 8dAR).
- Limited explanation/motivation for the thresholding methods (URNk, PhJW).
- Models used in the experiments are restricted to BAAI/OpenAI models (8dAR).
- Lack of discussion on prior work (PhJW).
- Lack of coherence in the narrative of the paper makes it difficult to find the unique contribution of the paper (PhJW, Ypbg).
- Limited novelty, as the benefits of domain-specific fine-tuning are already widely studied (8dAR, Ypbg).

**Additional Comments On Reviewer Discussion:**

Author provided responses but reviewers' concerns were not sufficiently resolved.

---

### Decision · Program_Chairs · 2025-01-22

Reject